# Accelerated photosynthesis routine in LPJmL4

**Jenny Niebsch[1], Werner von Bloh[2], Kirsten Thonicke[2], and Ronny Ramlau[1]**

[1]RICAM, Altenbergerstr. 69, 4040 Linz, Austria
[2]Potsdam Institute for Climate Impact Research (PIK), Member of the Leibniz Association, 14412 Potsdam, Germany

**Correspondence:** Jenny Niebsch (jenny.niebsch@oeaw.ac.at)

**Abstract.** The increasing impacts of climate change require strategies for climate adaptation. Dynamic global vegetation models (DGVMs) are one type of multi-sectorial impact model with which the effects of multiple interacting processes in the terrestrial biosphere under climate change can be studied. The complexity of DGVMs is increasing as more and more processes, especially for plant physiology, are implemented. Therefore, there is a growing demand for increasing the computational performance of the underlying algorithms as well as ensuring their numerical accuracy. One way to approach this issue is to analyse the routines which have the potential for improved computational efficiency and/or increased accuracy when applying sophisticated mathematical methods.

In this paper, the Farquhar–Collatz photosynthesis model under water stress as implemented in the Lund–Potsdam–Jena managed Land DGVM (4.0.002) was examined. We additionally tested the uncertainty of most important parameter of photosynthesis as an additional approach to improve model quality. We found that the numerical solution of a nonlinear equation, so far solved with the bisection method, could be significantly improved by using Newton's method instead. The latter requires the computation of the derivative of the underlying function which is presented. Model simulations show a significantly lower number of iterations to solve the equation numerically and an overall run time reduction of the model of about 16 % depending on the chosen accuracy. Increasing the parameters $\theta$ and $\alpha_{C_3}$ by 10 %, respectively, while keeping all other parameters at their original value, increased global gross primary production (GPP) by 2.384 and 9.542 GtC yr$^{-1}$, respectively. The Farquhar–Collatz photosynthesis model forms the core component in many DGVMs and land surface models. An update in the numerical solution of the nonlinear equation in connection with adjusting globally important parameters to best known values can therefore be applied to similar photosynthesis models. Furthermore, this exercise can serve as an example for improving computationally costly routines while improving their mathematical accuracy.

## 1 Introduction

Climate change is increasingly affecting the world we live in, and that in turn affects nature's contribution to our livelihoods (Pörtner et al., 2022). Estimating the extent and impact of climate change has become more and more urgent over the last couple of decades. Earth system models (ESMs) as well as impact models are used to develop strategies for climate adaptation and mitigation to achieve the Paris climate accord (Masson-Delmotte et al., 2021; Pörtner et al., 2022). Climate change affects vegetation dynamics, biodiversity, water, and biogeochemical cycles, which could reduce the biosphere's capacity to absorb carbon from the atmosphere in the future. Dynamic global vegetation models (DGVMs) are applied to study the net effects of multiple interacting processes that affect carbon sequestration (photosynthesis) and storage (in biomass and soil), see Prentice et al. (2007). It shows the demand for reliable and consistent model projections which require continuous work on reducing model uncertainty. While increasing complexity of the models by including more and more processes in DGVMs has been matched by increasing high-performance computing capabilities over the past decades, little has been invested into identifying and optimising computationally intensive routines in the model (Reichstein et al., 2019). These routines often have a long model history as they frequently belong to the core routines stemming from the very first model version. This includes, e.g.

the physiological modelling core of simulating photosynthesis in connection with atmospheric water demand or plant-water stress. The photosynthesis model is based on the Farquhar approach (Collatz et al., 1991, 1992; Farquhar et al., 1980) TS1 implemented in land surface schemes of the second generation (Pitman, 2003) followed by the first global biome models (Haxeltine and Prentice, 1996a) from which DGVMs evolved later on (Prentice et al., 2007).

The Farquhar–Collatz approach was implemented in the land surface of the SiB2 model by Sellers et al. (1992, 1996a), where it replaced their empirical photosynthesis model. The photosynthesis model in SiB2 (Sellers et al., 1996b) covers the co-limitation by Rubisco enzyme activity, light availability, and export limitation of carbon compounds. Furthermore, it covers the gradient between inner-stomatal $CO_2$ concentration to the $CO_2$ concentration around the leaf surface in the computation of stomatal conductance. By implementing the semi-mechanistic photosynthesis model and coupling it to transpiration via stomatal conductance, the land surface model (LSM) could then not only investigate biophysical effects of climate change but also the biogeochemical effects of rising atmospheric $CO_2$ in the earth system (Pitman, 2003). The SiB2 model (Sellers et al., 1992, 1996a), the NCAR CCM2 model (Bonan et al., 1995), and the MOSES land surface model of the UK Met office (Cox et al., 1998) were among the first to implement this photosynthesis scheme and evaluate it against field campaigns. At present, the Farquhar–Collatz photosynthesis model is used in a number of land surface models of the CMIP-5 earth system models, such as the Community Atmosphere Biosphere Land Exchange (CABLE), the LSM of the Australian community climate and earth system simulator (ACCESS, see de Kauwe et al., 2015, and ref. therein), as well as the ORCHIDEE DGVM (Krinner et al., 2005) of the IPSL-CM5 earth system model (Dufresne et al., 2013). Different models of stomatal conductance were evaluated for the JSBACH LSM (Reick et al., 2013) of the Max Planck Institute earth system model (MPI-ESM) to account for hydraulic properties and drought response (Knauer et al., 2015). The Community Land Model CLM4.5 (Oleson et al., 2013) of the NCAR ESM use the Ball–Berry model of stomatal conductance and extended it to account for leaf temperature acclimation and leaf water potential (Bonan et al., 2014); a similar approach was implemented in the JULES-vn5.6 land surface model (Oliver et al., 2022) of the UK Hadley Centre ESM (Sellar et al., 2019).

While land surface models detail vertical water, energy, and carbon profiles within the canopy, which extrapolates the photosynthetic capacity calculated at the leaf level to canopy photosynthesis (Sellers et al., 1996b), stand-alone DGVMs often use a big-leaf approach and compute daytime photosynthesis for canopy conductance, which goes back to the BIOME-3 model (Haxeltine and Prentice, 1996b) that opened up the second line of vegetation models by embedding the Farquhar–Collatz photosynthesis model in a mod-

elling framework of plant physiology and vegetation dynamics in DGVMs (Prentice et al., 2007). The Haxeltine and Prentice (1996b) implementation is used in the LPJ model family originating from Sitch et al. (2003) and the LPJ-GUESS model (Smith et al., 2001, 2014), as well as the current LPJmLv4 model (Schaphoff et al., 2018a, b). The Farquhar photosynthesis module forms the core of many other DGVMs, see e.g. Smith et al. (2001, 2014); Krinner et al. (2005). Today, 14 DGVMs (stand-alone and coupled to land surface models) contribute to the TRENDY intercomparison project (https://blogs.exeter.ac.uk/trendy/, last access: 14 December 2022), which informs the global carbon project on the state of the land carbon sink (Sitch et al., 2015).

In order to apply the model to the global land surface it is no longer sufficient to use faster or larger computing infrastructure or to try to parallelise the code as in von Bloh et al. (2010). Rather it requires the evaluation of the underlying algorithm structure of the code and, in particular, the used numerical methods. Replacing "old" numerical algorithms with modern methods will result in a significantly better run time performance while simultaneously maintaining or even increasing the accuracy of the method. We quantified the run time required by each submodule (or routine) of the LPJmL DGVM using the profiling option of the compilation command and the Linux gprof utility. We found that the repeated execution of the photosynthesis routine demands a big fraction, i.e. 38 %, of the computational time. All other routines require less than 11 %.

To illustrate our approach, our goal was to improve the computational efficiency of DGVMs by accelerating the photosynthesis module under water stress conditions using the Lund–Potsdam–Jena DGVM, Schaphoff et al. (2018a, b) as an example. A key ingredient in the modelling of photosynthesis is the determination of the ratio $\lambda$ between intracellular and ambient $CO_2$ concentration. Mathematically, $\lambda$ is computed as a zero of a nonlinear equation $f(\lambda) = 0$, which has so far been solved by a simple bisection algorithm. We expected to improve the computational efficiency by applying one of the more sophisticated solution methods, namely Regula falsi, secant and Newton's method. In this technical paper, we describe testing all three methods but found that only with Newton's method was the computational efficiency significantly improved. Only a few detailed specialised studies mention the use of Newton's or similar methods to solve coupled balance schemes, (Collatz et al., 1991; Pearcy et al., 1997; Soo-Hyung and Lieth, 2003; Dubois et al., 2007), or extensions of the photosynthesis-transpiration scheme along the leaf–plant–soil continuum in DGVMs (Bonan et al., 2014) are mentioned, but none provide documentation on the computational efficiency or how the numerical method was implemented in the model and/or their code. In addition, we test the effect of sensitive photosynthesis parameters on the annual gross primary production (GPP) CE1 of the computationally efficient model where we build on recent work by Walker et al. (2020).

We start with a short description of the different mathematical methods to find the zeros of a general nonlinear continuous function $f$ and their advantages and disadvantages. Afterwards, we introduce the relevant function $f$ from the photosynthesis module and calculate its derivative. We then compare the performance of Newton's algorithm and bisection in terms of the number of iterations and the computational time that is necessary to achieve a given accuracy. Finally, we benchmark the updated LPJmL version to show that the simulated vegetation dynamics as well as storage and fluxes of carbon and water remain robust.

## 2 Solution of nonlinear equations

The computation of the ratio $\lambda$ between intracellular and ambient $CO_2$ concentrations requires us to compute the zero of a function $f(\lambda)$. In most cases, this task cannot be solved analytically but requires a numerical approach, mostly based on iterative methods. Given a nonlinear continuous function TS2 $f : \mathbb{R} \rightarrow \mathbb{R}$, we want to find the zero(s) $x_s$ of this function within a certain interval $[a, b]$. While bisection, regula falsi, and secant methods are very simple to implement, Newton's method requires the computation of the derivative of $f$, which will be provided for the photosynthesis equation described in Sect. 3.2.

Here, the computational efficiency is determined by the speed of convergence. To compare the methods with respect to the speed of convergence we define the order of convergence as follows: let $x_s$ be a zero of $f$ found by computing a sequence $(x_k)$ of approximate solutions via an iteration scheme. The iteration method has the order of convergence $p$ if

$$\limsup_{k \to \infty} \frac{\|x_{k+1} - x_s\|}{\|x_k - x_s\|^p} = K, \tag{1}$$

with $0 < K < \infty$ and $K < 1$ for $p = 1$. Thus a high order of convergence implies a fast convergence, which on the other hand means fewer iteration steps. Numerically, the iteration is stopped either if the function value $f(x_k)$ of the iterate $x_k$ is almost zero, i.e. less than a given accuracy $y_{acc}$, or if the iterate itself changes less than a given accuracy $|x_k - x_{k-1}| < x_{acc}$.

Let us introduce some of the methods in the following subsections, see Schwarz and Köckler (2009) for details.

### 2.1 Bisection

For bisection we have to choose $[a, b]$ such that $f(a) \cdot f(b) < 0$, i.e. $f(a)$ and $f(b)$ have different signs. We compute the midpoint of the interval $x_m = \frac{a+b}{2}$ and its function value $f(x_m)$. If $|f(x_m)| < y_{acc}$ the search is complete, if not we check if $f(a) \cdot f(x_m) < 0$. If the latter is the case, $x_s$ has to be in the interval $[a, x_m]$ or otherwise in $[x_m, b]$. We repeat this bisection until either $|f(x_k)| < y_{acc}$ or $|x_k - x_{k-1}| < x_{acc}$.

This method always converges but slowly with convergence order $p = 1$, i.e. linear convergence.

### 2.2 Regula falsi

For the regula falsi method, we also need to choose $a, b$ such that $f(a) \cdot f(b) < 0$. Instead of the midpoint of $[a, b]$, we compute the next iterate $x_1$ for an approximation of $x_s$ by computing the zero of the linear function through the points $(a | f(a))$ and $(b | f(b))$. Again we check if $|f(x_1)| < y_{acc}$ and abort or check if $f(a) \cdot f(x_1) < 0$, and repeat this procedure either with $[a, x_1]$ or $[x_1, b]$. Convergence is always assured and is also linear, i.e. $p = 1$.

### 2.3 Secant method

The secant method only differs from the regula falsi in that the starting values $a = x_0$ and $b = x_1$ do not have to fulfil the condition $f(a) \cdot f(b) < 0$. The next iterate is computed by

$$x_{k+1} = x_k - f(x_k) \frac{x_k - x_{k-1}}{f(x_k) - f(x_{k-1})}. \tag{2}$$

This method can fail to converge depending on the starting values. If the method converges, it does so with order $p = 1.618$ TS3. Since the conditions on the starting values to ensure convergence depend on the knowledge of $x_s$, in practise $a$ and $b$ still have to fulfil the condition $f(a) \cdot f(b) < 0$.

### 2.4 Newton's method

Newton's method starts at an arbitrary approximation $x_0$ of $x_s$ and uses the tangent of the function $f$ at $(x_0, f(x_0))$ to compute the next iterate $x_1$ as the zero of the tangent. This is repeated, thus the next iterate is always computed from the previous one by

$$x_{k+1} = x_k - \frac{f(x_k)}{f'(x_k)}, \tag{3}$$

provided that $f'(x_k) \neq 0$. The method belongs to the class of fixed point iterations because the computation of the next iterate depends on the previous iterate only. If $f$ is three times differentiable on $[a, b]$ and $f'(x_s) \neq 0$, then there exists an interval $I = [x_s - \delta, x_s + \delta]$ such that $f$ is a contraction on $I$. It implies that for every start value from $I$, the method converges at least with order $p = 2$ (Schwarz and Köckler, 2009). We remark that the gain in convergence speed has to be weighted against the time it takes to compute the derivative of $f$.

## 3 Application to the problem

We now analyse the difference in speed of convergence between the bisection and Newton's methods when applied to the optimisation equation of the photosynthesis routine of the LPJmL DGVM.

## 3.1 Definition of the function $f$

In presenting the function $f(\lambda)$, we follow the nomenclature of Schaphoff et al. (2018a), which contains a detailed description of the derivation of this function. A list of the used symbols is given in Appendix A. We want to find $\lambda = \frac{c_i}{c_a} = \frac{p_i}{p_a}$, i.e. the ratio between the intracellular and ambient $CO_2$ concentration, or partial pressure, respectively, as the solution of the following equation:

$$0 = f(\lambda) = A_{nd}(\lambda) + \left(1 - \frac{dayl}{24}\right) R_{leaf}$$
$$- \frac{p_a(g_c - g_{min})}{1.6}(1 - \lambda). \qquad (4)$$

Here $A_{nd}$ is the net daily photosynthesis, $R_{leaf}$ is the leaf respiration, dayl is the hours of daylight, $p_a$ is the ambient partial pressure, $g_c$ is the canopy conductance, and $g_{min}$ is the minimum canopy conductance for a specific plant functional type (PFT). The first term is the photosynthesis during daylight. It is the gross daily photosynthesis $A_{gd}$ minus leaf respiration, $A_{nd}(\lambda) = A_{gd}(\lambda) - R_{leaf}$. The second term represents the dark respiration, i.e. respiration during night time. The third term represents the photosynthesis that is possible to achieve a potential canopy conductance. In finding $\lambda$ such that $f(\lambda) \approx 0$ we actually balance both light- and Rubisco-limited photosynthesis (first two terms) and photosynthesis related to the potential canopy conductance.

To shorten the formulas we define the abbreviation $C_{pg} = \frac{p_a(g_c - g_{min})}{1.6}$ as

$$0 = f(\lambda) = A_{gd}(\lambda) - \frac{dayl}{24} R_{leaf} - C_{pg}(1 - \lambda). \qquad (5)$$

The second summand does not depend on $\lambda$, whereas $A_{gd}(\lambda)$ has a more complex representation. The gross photosynthesis rate $A_g$ is the minimum of the light-limited, $J_C$, and Rubisco-limited photosynthesis rate, $J_E$. It can be shown that the minimum can be computed as

$$A_{gd}(\lambda) = \frac{dayl}{2\theta} [J_E(\lambda) + J_C(\lambda)$$
$$- \sqrt{(J_E(\lambda) + J_C(\lambda))^2 - 4\theta J_E(\lambda) J_C(\lambda)}], \qquad (6)$$

where $\theta$ is a shape parameter that allows for a gradual transition from one limitation to the other.

Light-limited photosynthesis depends on the absorbed photosynthetically active radiation (APAR); Rubisco-limited photosynthesis is determined by the maximum Rubisco capacity $V_m$:

$$J_E(\lambda) = C_1(\lambda) \frac{APAR}{dayl}, \qquad (7)$$

$$J_C(\lambda) = C_2(\lambda) V_m. \qquad (8)$$

Setting the internal partial pressure $p_i = \lambda p_a$ and using another abbreviation $C_K = K_c(1 + \frac{[O_2]}{K_O})$, where $K_c$ is the

Michaelis constant for $CO_2$ and $[O_2]$ and $K_O$ are the partial pressure and the Michaelis constant for oxygen, we have

$$C_1(\lambda) = \begin{cases} T_{stress}\, \alpha_{C_3}\, \frac{\lambda p_a - \Gamma_*}{\lambda p_a + (2)\Gamma_*} & \text{for } C_3\text{-Photosynthesis} \\ T_{stress}\, \alpha_{C_4}\, \frac{\lambda}{\lambda_{max_{C_4}}} & \text{for } C_4\text{-Photosynthesis.} \end{cases} \qquad (9)$$

$$C_2(\lambda) = \begin{cases} \frac{\lambda p_a - \Gamma_*}{\lambda p_a + C_K} & \text{for } C_3\text{-Photosynthesis} \\ 1 & \text{for } C_4\text{-Photosynthesis.} \end{cases} \qquad (10)$$

Here, $\alpha_{C_3}$ and $\alpha_{C_4}$ are the intrinsic quantum efficiencies for $CO_2$ uptake in $C_3$ and $C_4$ plants, respectively. $\Gamma_*$ is the carbon dioxide compensation point and $T_{stress}$ is a temperature stress function defined as

$$T_{stress} = \frac{1 - 0.01 e^{T_3(T_d - T_4)}}{1 + e^{T_1(T_2 - T_d)}}, \qquad (11)$$

with $T_d$ as the daily air temperature and $T_1$ to $T_4$ being PFT-specific temperature parameters (Sitch et al., 2000). LPJmL simulates vegetation dynamics for the 10 PFTs; we provide the parameter values used for $T_1$ to $T_4$ in Appendix A, Table A1, for the PFT types from Schaphoff et al. (2018a).

## 3.2 Derivative of $f$

To compute the derivative $f'$ of $f$ we rearrange Eq. (5):

$$f(\lambda) = A_{gd}(\lambda) + C_{pg}\lambda - C_{pg} - \frac{dayl}{24} R_{leaf}. \qquad (12)$$

Since the last two terms are constant the derivative is given by

$$f'(\lambda) = A'_{gd}(\lambda) + C_{pg}. \qquad (13)$$

To determine $A'_{gd}$ we apply sum, chain, and product rule of differentiation to Eq. (6) and get

$$A'_{gd}(\lambda) = \frac{dayl}{2\theta}$$
$$\times \left[ J'_E + J'_C - \frac{[J_E + J_C][J'_E + J'_C] - 2\theta[J'_E J_C + J_E J'_C]}{\sqrt{(J_E + J_C)^2 - 4\theta J_E J_C}} \right]. \qquad (14)$$

The derivatives of $J_E$ and $J_C$ are given by

$$J'_E(\lambda) = C'_1(\lambda) \frac{APAR}{dayl}, \qquad (15)$$

$$J'_C(\lambda) = C'_2(\lambda) V_m. \qquad (16)$$

To compute $C'_1$ from Eq. (9) and $C'_2$ from Eq. (10) we use the quotient rule

$$C'_1(\lambda) = \begin{cases} T_{stress}\, \alpha_{C_3}\, \frac{2(3)p_a\Gamma_*}{(\lambda p_a + (2)\Gamma_*)^2} & \text{for } C_3\text{-Photosynthesis} \\ \frac{T_{stress}\, \alpha_{C_4}}{\lambda_{max_{C_4}}} & \text{for } C_4\text{-Photosynthesis.} \end{cases} \qquad (17)$$

$$C'_2(\lambda) = \begin{cases} \frac{p_a(C_K + \Gamma_*)}{(\lambda p_a + C_K)^2} & \text{for } C_3\text{-Photosynthesis} \\ 0 & \text{for } C_4\text{-Photosynthesis.} \end{cases} \qquad (18)$$

We describe the consequent changes in the model code which were required to implement the computation of the derivative fcnd($\lambda$) in the Appendix B.

The function $f$ is defined for all $\lambda > 0$, as long as $(J_E(\lambda) + J_C(\lambda))^2 \geq 4\theta J_E(\lambda) J_C(\lambda)$. As a composition of at least 3 times differentiable functions, it fulfils the differentiability condition of Newton's method. The parameters in the definition of $f$ vary with the geographic location and season. A plot of $f$ for parameters from different locations (boreal, temperate, and tropical) and at different times can be seen in Fig. 1.

The condition $f'(\lambda) \neq 0$ as well as the suitability of a staring value can not be generally ensured. In all our computations convergence was not a problem. To be on the safe side, one can implement a hybrid method that switches to bisection if convergence of the iterates does not occur.

## 4 Numerical performance and discussion

We have tested the different methods in the routine regarding computational time and number of iterations for given accuracy $x_{\mathrm{acc}}$. There was no significant speed up with the secant and regula falsi method. Hence, we concentrated on the comparison of bisection and Newton's methods and describe the outcome in this section.

In a first test, the LPJmL model was run over 120 simulation years and the number of iterations in the bisection and Newton's routine was counted and averaged over all grid cells and one year (Fig. 2). For $x_{\mathrm{acc}} = 0.01$ this number was about 3 for Newton's method and 7 for bisection (dotted lines in Fig. 2). When $x_{\mathrm{acc}}$ was set to 0.001 the number of iterations with Newton's method increased only slightly, whereas the bisection method needed 9 to 10 iterations (solid lines in Fig. 2). Until now, the bisection algorithm used 10 as the maximal number of iterations. Using maximum 10 iterations fits into the interval width of $2^{-10} \approx 0.001$, our accuracy measure $x_{\mathrm{acc}}$. Increasing the maximum number of iterations had no effect on the number of required iterations. We conclude that Newton's method reduces the necessary number of iteration to a third.

In a next step, a spin-up run of LPJmL over 5000 simulation years was conducted to compare the time performance using both routines. Usually, LPJmL simulation experiments start from bare ground, i.e. initial vegetation conditions are not prescribed. Therefore, a spin-up run is used to bring all vegetation and soil carbon pools into equilibrium with climate. For the usually implemented accuracy $x_{\mathrm{acc}} = 0.1$ the computation time for 5000 years was about 5250 s in both cases. This means that the advantage of Newton's method in terms of iteration numbers is levelled by the additional time for computing the derivative of $f$. For $x_{\mathrm{acc}} = 0.01$, the bisection method needed 6700 s, while Newton's method needed 5600 s. Thus a reduction of about 16 % in time could be observed. It implies that with almost the

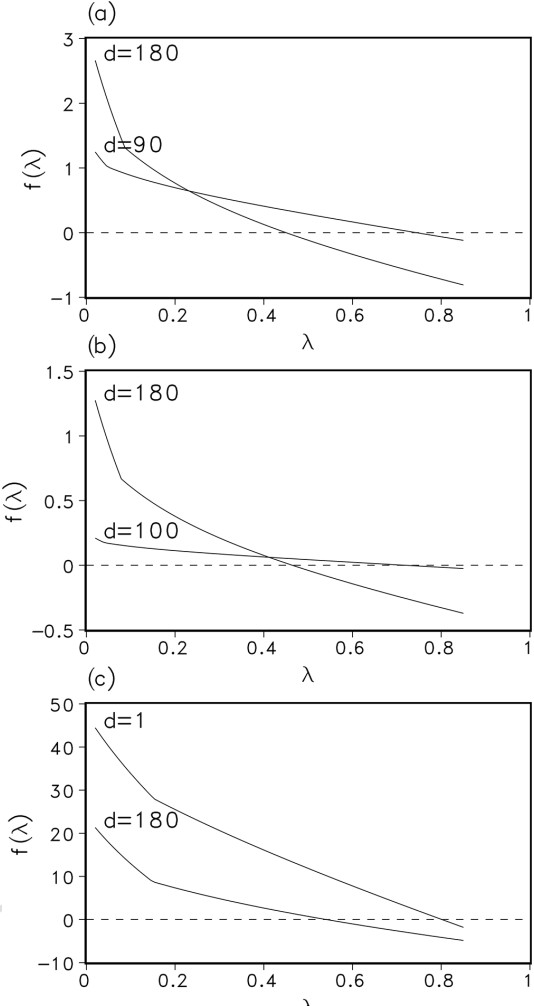

**Figure 1.** Function $f(\lambda)$ for a set of parameters from different days in 1901 and locations, namely Hainich (Germany, mixed-temperate forest; **(a)**, Seiteminen (Finland, boreal forest; **(b)**, and Santarem (Brazil, tropical rainforest; **(c)**. Panel **(d)** denotes the day in year 1901.

same amount of time (5250 s vs. 5600 s) a higher accuracy can be achieved with Newton's method (Fig. 3). While the accuracy $y_{\mathrm{acc}}$ does not increase significantly for the bisection method for $x_{\mathrm{acc}} = 0.001$, we gain a 2 orders of magnitude increase in $y_{\mathrm{acc}}$ for the Newton's method. As a result, a change of $x_{\mathrm{acc}}$ from 0.1 to 0.01 will be permanently implemented in the LPJmL model for future model applications. We expect that with the implementation of new model developments that affect the photosynthesis module (e.g. nutrient limitation from nitrogen and leaf temperatures) an efficient and increased model accuracy ($y_{\mathrm{acc}}$) for finding the zero of $f(\lambda)$ will be even more important. It can be expected that the computation time for the bisection method would increase substantially, while increasing only moderately for Newton's method.

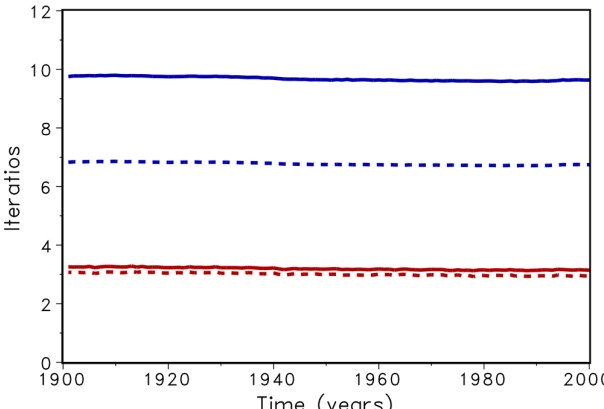

**Figure 2.** Average number of iteration for bisection (upper lines, blue) and Newton (lower lines, red) for accuracy $x_{acc} = 0.01$ (dotted) and 0.001 (solid).

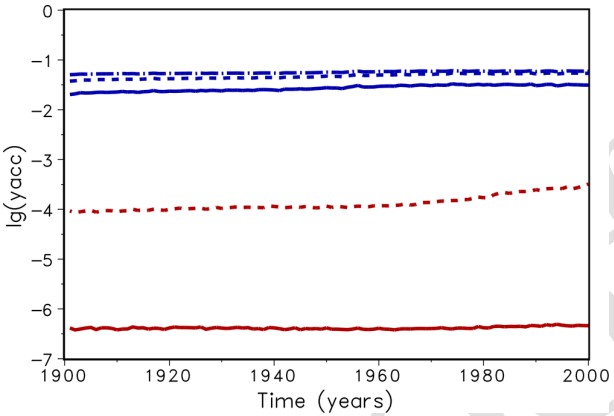

**Figure 3.** Mean decadic logarithm of the accuracy $y_{acc}$ for bisection (upper lines, blue) and Newton (lower lines, red) for accuracy $x_{acc} = 0.01$ (dotted) and 0.001 (solid). The dashed–dotted line shows the accuracy of the original version of LPJmL.

In order to check if the implementation of Newton's method is robust for all important model variables, we performed a transient simulation with the LPJmL model starting from the spin-up and covering the years 1901–2000. Model configuration and input data are as in Schaphoff et al. (2018a). We compared the main diagnostic variables of the published LPJmL4.0 version against the version using Newton's method (see Appendix C). We found that most global diagnostic variables related to fluxes and storage of carbon and water had differences of $< \pm 1.0\%$, including total vegetated area. Only marginal changes ($+3$ gC per mA$^2$ TS4 and month) in net primary productivity (NPP), heterotrophic respiration, and evaporation are seen mainly in Europe and southern as well as southeastern Asia. The reductions in carbon storage in litter and soil are very small and apply only to the boreal zone across the Northern Hemisphere and central

Europe (compare spatial maps of carbon and water variables in Appendix C).

The photosynthesis module is also applied to the crop functional types and managed grassland within LPJmL4.0. Therefore, sawing dates, crop productivity, and harvest are among the simulated variables. Comparing both model versions in the model benchmark, we found that global harvest changed for a number of crops. Rainfed and irrigated rice increased by 5% and 8%, respectively, mainly in India and southeast Asia. Harvest of rainfed temperate cereals increased by $< 1.0\%$, mainly found in central Europe. Harvest of irrigated temperate cereals (incl. wheat) increased by 4.5%, which mainly applied to India as well. Harvest of irrigated and rainfed soybeans increased by 2.3% and 1.5% globally; the differences are mainly found in the US and Brazil. All other crop functional types had marginal to zero changes in global productivity as well as simulated harvest (see Table in Appendix C).

For all global carbon pools (vegetation and soil) and carbon (GPP, heterotrophic respiration, and fire emissions) as well as water fluxes (transpiration and runoff) we found no difference in the temporal changes in the transient simulation over the 20th century. All variables showed similar, if not identical, dynamics (data not shown). Small changes were found in the fractional coverage of plant functional types, i.e. most differences were negligible. The fractional coverage of temperate broadleaved summergreen trees increased by 4.8% globally, which mainly applies to Europe, the northeastern USA, and parts of China. Increases in temperate $C_3$ grasses are found in the boreal zone, summing up to 4.8% globally. Marginal changes of $< 0.5\%$ per grid cell are found for all other PFTs, which imply small adjustments in vegetation composition in these vegetation zones (see difference maps in Appendix C). Comparisons using flux tower measurements on carbon and water fluxes as well as discharge data showed no differences so we can conclude that also for these variables the results are robust (data not shown). We can therefore conclude that the LPJmL results were robust before but are now achieved due to improved accuracy of the photosynthesis routine.

After improving the computational efficiency and numerical precision, we can now test the parameter uncertainties following Walker et al. (2020), who tested the sensitivity of $\theta$, $\alpha_{C_3}$, $b_{C_3}$, $k_{c25}$, and $K_{o25}$ on their impacts on global GPP. The LPJmL model computes $V_m$ as follows (Schaphoff et al., 2018a, Eq. 35):

$$V_m = \frac{1}{b_{C_3}} \cdot \frac{c_1}{c_2} \cdot ((2\theta - 1) \times s - (2\theta \times s - c_2) \times \sigma) \cdot \text{APAR}. \quad (19)$$

Therefore, the sensitivity of $V_{cmax}$ results from varying $b_{C_3}$ indirectly since the reciprocal of $b_{C_3}$ is used to calculate $V_{cmax}$ in a linear equation. Varying $b_{C_3}$ is therefore the adequate sensitivity test which relates to $V_{cmax}$. We varied each parameter by 10% independently and find that $\theta$ ($\alpha_{C_3}$, $b_{C_3}$, $k_{c25}$, $K_{o25}$) increases global annual GPP (AGPP, hereafter)

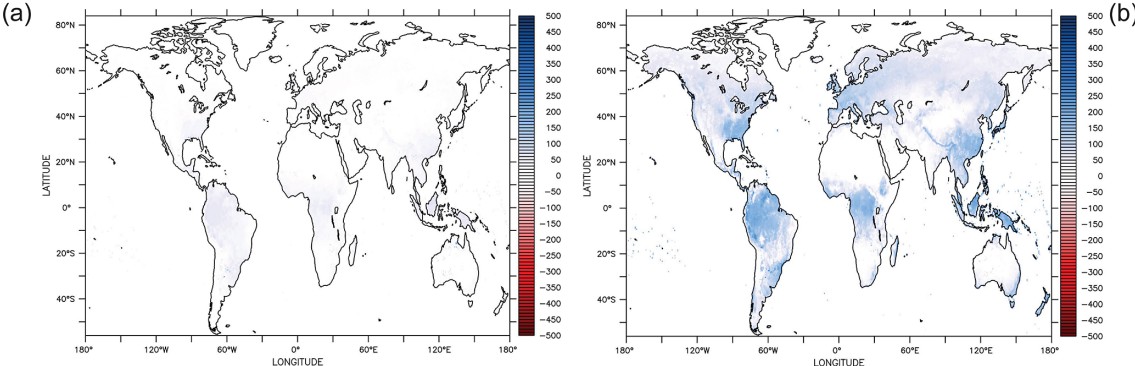

**Figure 4.** Parameter sensitivity on annual gross primary productivity (AGPP, average of 1901–2000) shown as the difference between new parameter and reference simulations. Both simulations have the Newton approach implemented. Increasing $\theta$ by 10 % increased AGPP mainly in forested regions (**a**). Increasing $\alpha_{C_3}$ by 10 % has a much larger effect on AGPP, especially in the tropics (**b**).

**Table 1.** Change in the AGPP after varying the listed parameters by 10 %. GPP is calculated as the global average mean for the years 1901–2000.

| Parameter | $\Delta$ GPP relative in % | $\Delta$ GPP absolute (GtC yr$^{-1}$) |
|---|---|---|
| $\theta$ | 1.67 | 2.384 |
| $\alpha_{C_3}$ | 6.68 | 9.542 |
| $b_{C_3}$ | −0.56 | −0.798 |
| $k_{c25}$ | −0.35 | −0.506 |
| $K_{o25}$ | 0.14 | 0.199 |

by 1.67 % (+6.69 %, −1.67 %, −0.35 %, and +0.14 %). Table 1 shows the difference of the two most important parameter on global AGPP.

Geographically, increasing $\theta$ yields higher AGPP mainly in the tropics and temperate forest regions, where AGPP increases up to $100\,\mathrm{gC\,m^{-2}}$. However, AGPP increases between 200 and $500\,\mathrm{gC\,m^{-2}}$ when changing $\alpha_{C_3}$, see Fig. 4. It turns out that AGPP is increased in all regions, where LPJmL simulates woody PFTs. Also here, the largest effects are seen in (sub-)tropical and temperate regions which span larger areas than the areas with increased AGPP as a result of varying $\theta$.

We remark that future work on the photosynthesis approach could focus on the new Johnson and Berry scheme (Johnson and Berry, 2021) with the advantage of calculating gas exchange and relying less on empirical coefficients.

# 5 Conclusions

The computational load of dynamic global vegetation models, caused by increased complexity of the modelling processes, has so far been counteracted by the high-performance computing systems used. However, more recently it has become clear that updates in computing infrastructure are not sufficient anymore. Consequently, we proposed to carefully evaluate the algorithmic structure of DGVMs and identify and update routines that can benefit from the use of modern mathematical methods. As a showcase, we investigated the photosynthesis model in the LPJmL DGVM. Specifically, we investigated the computation of the ratio $\lambda$ between intracellular and ambient $CO_2$, which is obtained as the zero of a function $f$. We proposed to replace the so far used bisection method with a Newton method, which is known to converge significantly faster. We carefully compared the model performance of the published LPJmL4.0 version with the version developed in this study and found that the model performance is robust. Using a more sophisticated mathematical method in the photosynthesis module allowed for a higher precision in the computation of $\lambda$ and resulted in slightly increased productivity in continental and mountainous areas. We think that the new results are more accurate than the previous version due to the higher accuracy of the Newton method visible in Fig. 3. With the currently implemented accuracy bounds, the run time of the model with the Newton routine implemented is about 16 % lower than the old version. This advantage will be much more prominent if the complexity of the model is further extended or if more accurate modelling results are required. Consequently, the Newton-based routine will be implemented in the LPJmL model. Additionally, we believe that the Newton method can also be applied to photosynthesis modules in other DGVMs and can increase model accuracy and/or computational efficiency.

## Appendix A: Parameters in photosynthesis

General parameters used in the photosynthesis routine. PFT is plant functional type.

| | |
|---|---|
| $A_{nd}$ | daily net photosynthesis |
| dayl | day length |
| $R_{leaf}$ | leaf respiration |
| $p_a$ | ambient partial pressure |
| $g_c$ | canopy conductance |
| $g_{min}$ | PFT-specific minimum canopy conductance |
| $A_{gd}$ | daily gross photosynthesis |
| $\theta$ | co-limitation (shape) parameter |
| $J_E$ | light-limited photosynthesis rate |
| $J_C$ | Rubisco-limited photosynthesis rate |
| APAR | absorbed photosynthetically active radiation |
| $V_m$ | maximum Rubisco capacity |
| $K_C$ | Michaelis constant for $CO_2$ |
| $[O_2]$ | $O_2$ partial pressure |
| $K_O$ | Michaelis constant for $O_2$ |
| $T_{stress}$ | temperature stress function limiting photosynthesis at low and high temperatures |
| $\alpha_{C_3}$ | intrinsic quantum efficiencies for $CO_2$ uptake in $C_3$ plants |
| $\alpha_{C_4}$ | intrinsic quantum efficiencies for $CO_2$ uptake in $C_4$ plants |
| $\Gamma_*$ | carbon dioxide compensation point |
| $\lambda_{maxC_4}$ | maximum ratio of intracellular to ambient $CO_2$ for $C_4$-photosynthesis |

**Table A1.** PFT-specific parameter for temperature stress function (Eq. 12) in °C. PFT types as in Schaphoff et al. (2018a).

| Plant functional type (PFT) | $T_1$ | $T_2$ | $T_3$ | $T_4$ |
|---|---|---|---|---|
| Tropical broadleaved evergreen tree | 2.0 | 25.0 | 30.0 | 55.0 |
| Tropical broadleaved raingreen tree | 2.0 | 25.0 | 30.0 | 55.0 |
| Temperate needle-leaved evergreen tree | −4.0 | 20.0 | 30.0 | 42.0 |
| Temperate broadleaved evergreen tree | −4.0 | 20.0 | 30.0 | 42.0 |
| Temperate broadleaved summergreen tree | −4.0 | 20.0 | 25.0 | 38.0 |
| Boreal needle-leaved evergreen tree | −4.0 | 15.0 | 25.0 | 38.0 |
| Boreal needle-leaved summergreen tree | −4.0 | 15.0 | 25.0 | 38.0 |
| Polar $C_3$ grass | −4.0 | 10.0 | 30.0 | 45.0 |
| Temperate $C_3$ grass | −4.0 | 10.0 | 30.0 | 45.0 |
| Tropical $C_4$ grass | 6.0 | 20.0 | 45.0 | 55.0 |

## Appendix B: Programming

To implement Newton's method in the LPJmL code, changes had to be made in the functions `photosynthesis.c`, `gp_sum.c`, and `water_stressed.c` (separate file)

5 New function `newton.c`: see source code in a separate file.

*Remark.* The function `photosynthesis.c` within LPJmL computes the value $A_{\mathrm{nd}}(\lambda) + \left(1 - \frac{\mathrm{dayl}}{24}\right) R_{\mathrm{leaf}}$ for a given $\lambda$. In the function `water_stressed.c` the

10 function `fcn`$(\lambda)$ is defined as $\mathrm{fcn}(\lambda) = C_{\mathrm{pg}} \times (1 - \lambda) -$ photosythesis$(\lambda)$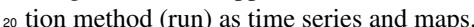, i.e. $\mathrm{fcn} = -f$. In order to use Newton's method we have to compute not only $\mathrm{fcn}(\lambda)$ but also its derivative $\mathrm{fcnd}(\lambda) = -f'(\lambda)$.

## Appendix C: LPJmL v4 benchmark results

15 The TS6 benchmark table of global status variables (Table C1) TS7 compares two model versions against each other and to literature values were available. The following Figs. D1–D6 show globally important variables simulated using the Newton approach (benchmark run) and the bisec-

20 tion method (run) as time series and maps.

**Table C1.** Global sums of actual vegetation, including land-use, comparing Newton approach (benchmark run) against bisection approach (run). Tece is temperate cereals. NA – not applicable, Mha – megahectare, Mt DM – megatonnes of dry matter. CE2

| Parameter | Lit. estimates | Run | Benchmark fun | Diff. abs. | Diff [%] |
|---|---|---|---|---|---|
| Vegetation carbon [GtC] | 460–660[a,b,c] | 595.9 | 596.2 | 0.231 | 0.039 |
| Total soil carbon density [GtC] | 2376–2456[d], 1567[e] TS8, 1395[f] TS9 | 1862 | 1862 | −0.08 | −0.004 |
| Litter carbon [GtC] | NA | 151.3 | 151.4 | 0.116 | 0.077 |
| Fire carbon emission [GtC yr$^{-1}$] | 2.14 (1.6 Nat. Fire)[g,h,i,j] | 3.108 | 3.109 | 0.001 | 0.036 |
| Establishment flux [GtC yr$^{-1}$] | NA | 0.161 | 0.161 | 0 | −0.002 |
| Area all natural vegetation [Mio ha] | NA | 7767 | 7767 | −0.119 | −0.002 |
| Area CE3 tropical broadleaved evergreen tree [Mio ha] | NA | 1180 | 1179 | −0.237 | −0.02 |
| Area tropical broadleaved raingreen tree [Mio ha] | NA | 1280 | 1280 | 0.448 | 0.035 |
| Area temperate needle-leaved evergreen tree [Mio ha] | NA | 364 | 360.8 | −3.166 | −0.87 |
| Area Temperate broadleaved evergreen tree [Mio ha] | NA | 322 | 321.5 | −0.467 | −0.145 |
| Area Temperate broadleaved summergreen tree [Mio ha] | NA | 136 | 142.5 | 6.517 | 4.792 |
| Area boreal needle-leaved evergreen tree [Mio ha] | NA | 429.2 | 426.8 | −2.393 | −0.558 |
| Area boreal broadleaved summergreen tree [Mio ha] | NA | 916.8 | 919.6 | 2.814 | 0.307 |
| Area boreal needle-leaved summergreen tree [Mio ha] | NA | 378.3 | 380.7 | 2.398 | 0.634 |
| Area tropical $C_4$ grass [Mio ha] | NA | 893.2 | 890.6 | −2.573 | −0.288 |
| Area temperate $C_3$ grass [Mio ha] | NA | 535.7 | 545.2 | 9.472 | 1.768 |
| Area polar $C_3$ grass [M] TS10 | NA | 1332 | 1320 | −12.93 | −0.971 |
| NPP [GtC yr$^{-1}$] | 66.05[k] TS11, 62.6[b], 49.52–59.74[l] | 62.81 | 62.87 | 0.064 | 0.102 |
| Heterotrophic respiration [GtC yr$^{-1}$] | NA | 50.78 | 50.83 | 0.044 | 0.086 |
| Evaporation [10 TS12 km$^3$ yr$^{-1}$] | NA | 9.644 | 9.661 | 0.017 | 0.173 |
| Transpiration [10 TS13 km$^3$ yr$^{-1}$] | NA | 47.83 | 47.82 | −0.011 | −0.024 |
| Interception [10 TS14 km$^3$ yr$^{-1}$] | NA | 7.914 | 7.912 | −0.002 | −0.024 |
| Runoff [10 TS15 km$^3$ yr$^{-1}$] | NA | 54.3 | 54.23 | −0.064 | −0.118 |
| Harvested carbon rainfed tece [Mt DM yr$^{-1}$ TS16] | 524.08[m] | 458.5 | 462.6 | 4.106 | 0.895 |
| Harvested carbon rainfed rice [Mt DM yr$^{-1}$] | 492.66[m] | 125.2 | 131.5 | 6.304 | 5.035 |
| Harvested carbon rainfed maize [Mt DM yr$^{-1}$] | 498.33[m] | 434.9 | 434.8 | −0.07 | −0.016 |

| Parameter | Lit. estimates | Run | Benchmark run | Diff. abs. | Diff [%] |
|---|---|---|---|---|---|
| Harvested carbon rainfed soybean [Mt DM yr$^{-1}$] | NA | 126.3 | 128.1 | 1.87 | 1.481 |
| Harvested carbon irrigated tece [Mt DM yr$^{-1}$] | 524.08[m] | 156.7 | 163.7 | 7.038 | 4.493 |
| Harvested carbon irrigated rice [Mt DM yr$^{-1}$] | 492.66[m] | 206.4 | 223 | 16.64 | 8.062 |
| Harvested carbon irrigated maize [Mt DM yr$^{-1}$] | 498.33[m] | 153.1 | 153.1 | −0.002 | −0.001 |
| Harvested carbon irrigated soybean [Mt DM yr$^{-1}$] | NA | 12.03 | 12.3 | 0.268 | 2.229 |
| tree cover fraction [–] | NA | 0.644 | 0.645 | 0.001 | 0.12 |

Literature: [a] Olson et al. (1985). [b] Saugier et al. (2001). [c] WBGU (1998). [d] Batjes (1996). [e] Eswaran et al. (1993). [f] Post et al. (1982). [g] Seiler and Crutzen (1980). [h] Andreae and Merlet (2001). [i] Ito and Penner (2004). [j] van der Werf et al. (2004). [k] Vitousek et al. (1986). [l] Ramakrishna et al. (2003). [m] FAOSTAT (2009).

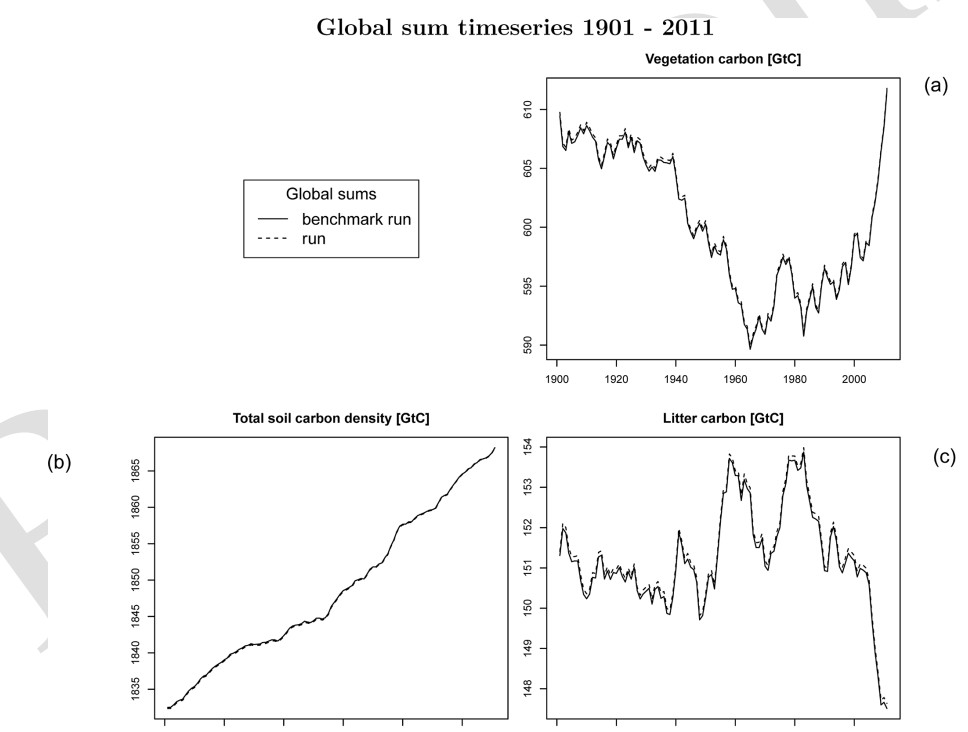

**Figure C1.** Global number for **(a)** vegetation carbon, **(b)** total soil carbon, and **(c)** litter carbon.

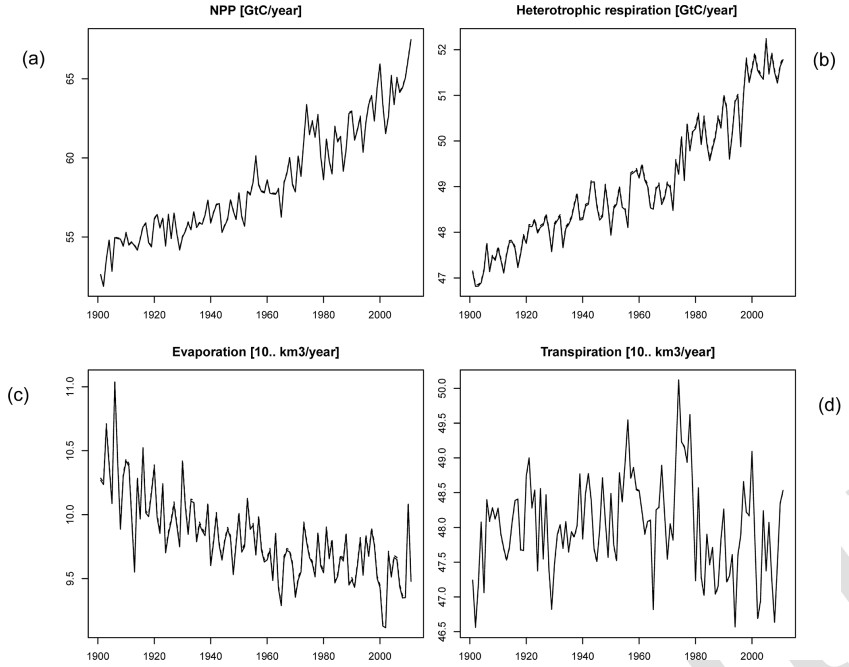

**Figure C2.** Global number for time series of **(a)** NPP, **(b)** heterotrophic respiration, **(c)** evaporation, and **(d)** transpiration.

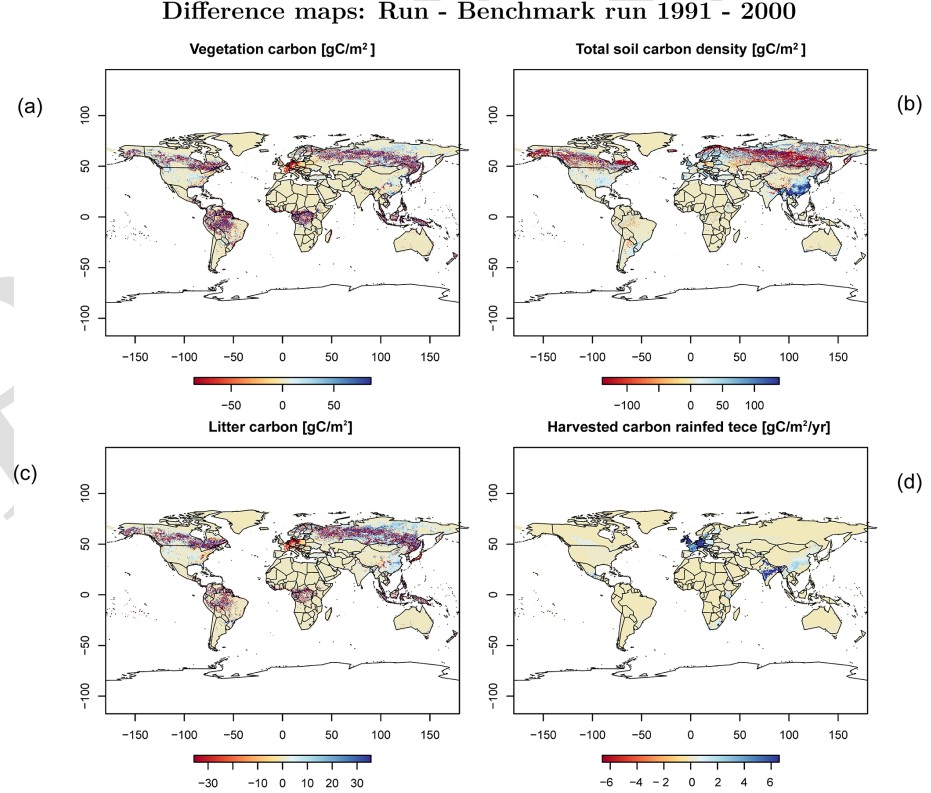

**Figure C3.** Difference maps of **(a)** vegetation carbon, **(b)** soil carbon, **(c)** litter carbon, and **(d)** harvested carbon of rainfed temperate cereals (tece).

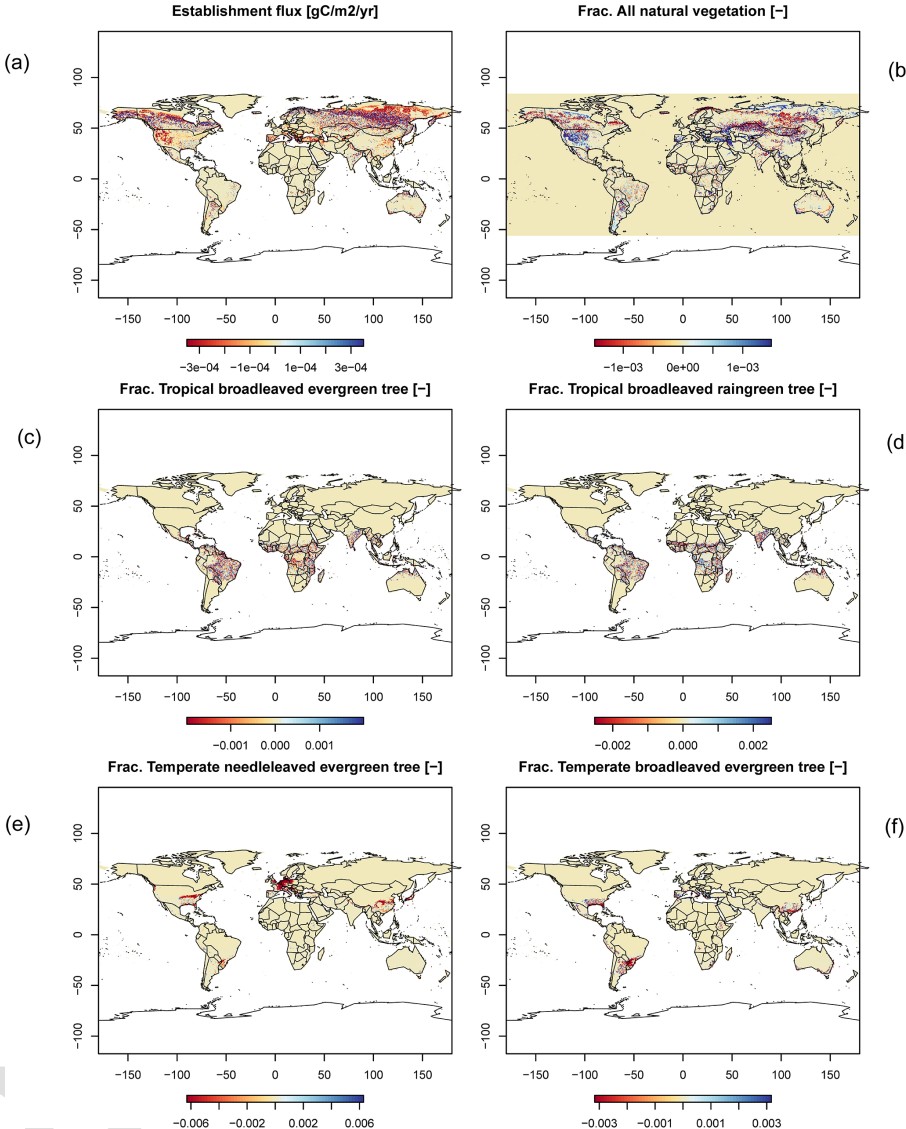

**Figure C4.** Difference maps of **(a)** establishment, **(b)** all natural vegetation, **(c)** frac. tropical broadleaved evergreen, **(d)** frac. tropical broadleaved raingreen, **(e)** frac. temperate needle-leaved evergreen, and **(f)** frac. temperate broadleaved evergreen.

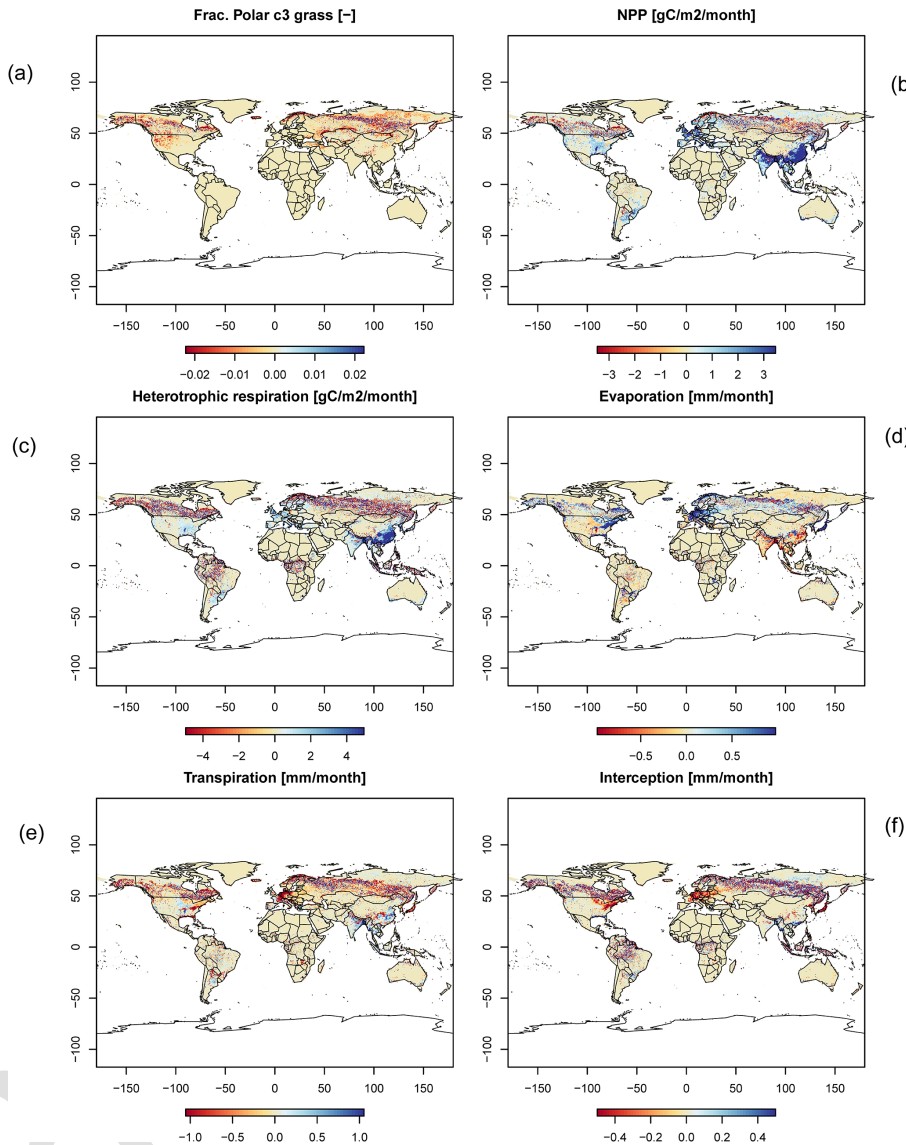

**Figure C5.** Difference maps of **(a)** frac. polar $C_3$ grass, **(b)** NPP, **(c)** heterotrophic respiration, **(d)** evaporation, **(e)** transpiration, and **(f)** interception.

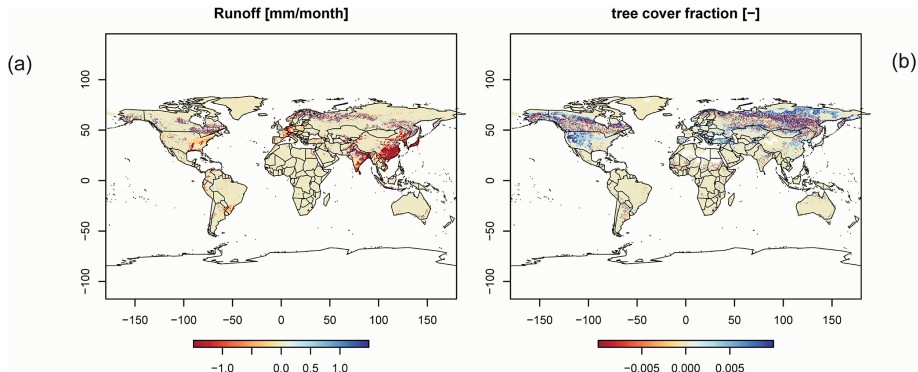

**Figure C6.** Difference maps of **(a)** runoff and **(b)** tree cover fraction.

*Code and data availability.* The model code is available at https: //doi.org/10.5281/zenodo.6644541 (Niebsch et al., 2022).

*Supplement.* The supplement related to this article is available online at: https://doi.org/10.5194/gmd-15-1-2022-supplement.

*Author contributions.* JN and RR performed the mathematical analysis, JN and WvB implemented and tested the new numerical methods, and WvB conducted the simulation experiments and analysed the model performance and computation efficiency. JN and KT wrote the paper and all authors contributed to the writing of the paper and discussion of the model study throughout to develop the work.

*Competing interests.* The contact author has declared that none of the authors has any competing interests.

*Acknowledgements.* The authors gratefully acknowledge the European Regional Development Fund (ERDF), the German Federal Ministry of Education and Research, and the Land Brandenburg for supporting this project by providing resources on the high-performance computer system at the Potsdam Institute for Climate Impact Research. We thank Marie Hemmen from PIK for her support in benchmarking the LPJmL model.

*Financial support.* . TS17

*Review statement.* This paper was edited by Carlos Sierra and reviewed by two anonymous referees.

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

**Remarks from the language copy-editor**

**Remarks from the typesetter**