# Peer review of "Accelerated photosynthesis routine in LPJmL4"

_Geoscientific Model Development, 2022_

## Author Comment (AC1)

**Autors response to referee 1**

November 7, 2022

We thank reviewer 1 for his/her constructive comments on our technical note. We provide a detailed response in the text below, where the R1 comments are marked in blue, our response in black. Changes in the revised manuscript are in magenta for referee 1 and in blue for referee 2.

Comment: In their paper "Accelerated photosynthesis routine in LPJmL4" the authors show that using a different algorithm in a subroutine of the photosyntheis computation leads to model speed up and higher numerical accuracy of the DGVM LPJmL. I very much agree with the authors that DGVMs need improvements in their numerical methods to decrease their computing time. Therefore, I see the proposed methodology as an important step towards this goal.
Response: We are grateful that the reviewer supports our argument on improving numerical methods by which the computational time and numerical accuracy for key routines that form the core of Dynamic Global Vegetation Models are improved. Given the fact that first versions of DGVMs were published in the late 1990ies and early 2000s, it becomes necessary to revisit the methods of existing core routines from which many other modelled processes in the model depend.

Comment: However, I find that replacing the bisection method with the Newton method to find the root of a continuous function does not suffice for a technical paper. A short technical comment could be appropriate, but quite frankly I believe that this (nonetheless important) improvement of LPJmL should simply be mentioned in the release notes of a new release of LPJmL.
Response: We agree that the extend of the study shown here is comparably small in relation to other model development papers. However, we would like to stress that we use the implementation of the Newton method exemplarily to show and underline the necessity on how mathematical knowledge can be used to revisit and improve existing routines in models that are now continuously developed and applied in, e.g., climate change studies, for nearly two decades. Moreover, we think that these aspects do not receive enough attention in publications of larger model update papers, which serve different objectives.

Comment: I also find that important things are not sufficiently discussed, namely: 1. There are only two citations when mentioning that this representation of photosynthesis is used in the majority of DGVMs. There are also other representations of PS and more citations will underline the point that Farquhar-Collatz is really the most used one.
Response: We thank the reviewer for pointing out that the references are not complete. Reviewer 2 had made a similar remark. We therefore refer to our respective author response. (The additional text in the revised manuscript is in blue.)

2. It should at least be mentioned that the function f suffices all criteria for the Newton-method
Response: The Newton method requires that $f$ is at least three times differentiable and the

first derivative of $f$ at the iteration is not zero. We now explain this in the text (in magenta in the revised manuscript) and the sentence now reads:

The function $f$ is defined for all $\lambda > 0$, as long as $(J_E(\lambda) + J_C(\lambda))^2 \geq 4\theta J_E(\lambda) J_C(\lambda)$. As a composition of at least three times differentiable functions it fulfills the differentiability condition of Newton's method.

The condition $f'(\lambda) \neq 0$ as well as the suitability of a staring value can not be generally ensured. In all our computations convergence was not a problem. To be on the safe side, one can implement a hybrid method that switches to bisection if convergence of the iterates does not occur.

3. Actually also a plot of f would be interesting to see, at least for one particular set of parameters, to let the reader get an impression of how this function looks like.

Response: We thank the reviewer for this helpful remark, as we agree that this additional plot helps to enhance the understanding of the function. We have identified the parameters that define the function f. Since some of these parameters vary with geographic location (average climate conditions) and season we have plotted them for a boreal, temperate and tropical site. All three sites are used as a standard in our model benchmarking. We have plotted f in a new Figure 1 and added the following text:

The parameters in the definition of $f$ vary with the geographic location and season. A plot of $f$ for parameters from different locations (boreal, temperate, and tropical) and at different times can be seen in Figure 1.

4. The Newton-method may fail when the starting value is chosen too far away from the root, it is not discussed whether this could become a problem.

Response: We discussed this comment in the text (see above): The condition $f'(\lambda) \neq 0$ as well as the suitability of a staring value can not be generally ensured. In all our computations convergence was not a problem. To be on the safe side, one can implement a hybrid method that switches to bisection if convergence of the iterates does no occur.

5. Some outputs had much higher changes when the new method was applied. There is no discussion why that could be.

Response: We have stressed that the changes appear larger but are of small dimension due to small absolute values. Since these occur in areas of low productivity for which reliable validation data are difficult to obtain, therefore hard to decide which version yields more reliable results. We also refer to our detailed response in our response to Reviewer 2, where we additionally tested the parameter sensitivity on annual GPP.

Comment: To conclude, I unfortunately cannot recommend this manuscript for publication as I evaluate its impact as too low for a paper in GMD.

Response: We hope that we have provided the required information and additional explanations that the revised manuscript would deserve publication as a technical note in GMD, also with the material added following review 2.

---

## Author Comment (AC2)

**Autors response to referee 2**

November 7, 2022

We thank reviewer 2 for his/her constructive comments on our technical note, especially for the detailed suggestions and suggested references which helped us to substantially improve our manuscript. We provide a detailed response in the text below, where the comments from Reviewer 2 are marked in blue, our responses in black. Changes in the revised manuscript are in magenta for referee 1 and in blue for referee 2.

This topic is generally appropriate for a report in Geoscientific Model Development, but as currently written the manuscript is likely to be relatively low impact. Primary concerns are: (1) the application of Newton's method to this problem while logical is not novel; (2) while it speeds the solution, the marginal improvement is modest (only on the order of 16%); (3) the focus on the acceleration of the photosynthesis scheme overlooks substantial underlying problems with calibration and evaluation of this scheme. To increase the impact of this manuscript, I would suggest: (a) including a concise review of the numeric methods used to implement the Farquhar-Collatz style photosynthesis schemes in land surface models; (b) better contextualizing the importance of computational efficiency relative to other priorities for the development of the photosynthesis scheme; (c) condensing the figures down to one or two key visuals, summarizing the magnitude of the impact of Newton's method.

(a) re concise review. We thank the reviewer for this thoughtful suggestion. It allows us to reflect recent scientific discussions around the Farquhar-Collatz photosynthesis scheme in our manuscript. Please see our response to the specific point related to this general issue further below, where we describe the inserted literature review.

(b) re importance of computational efficiency vs. improvements of parameter: we thank the reviewer for this important suggestion. We have now tested the most sensitive parameters in the photosynthesis routine thanks to the work published in [Walker et al. 2020] and describe the outcome in comparison to the effect of improving computation efficiency in the manuscript. See our detailed reply further below.

(c) re condensing figures: We understand that the figures which form part of our standard benchmarking protocol of the LPJmL model to measure model improvements and consistency was misleading. We wanted to show that the model is still robust, although it did not improve the simulation of general model variables such as carbon storage and fluxes. See also our response below regarding detailed evaluation of the photosynthesis scheme and parameter sensitivity.

Line 30, The current text should be updated to accurately describe the pathway that the Farquhar-style model took into large-scale applications. The Farquhar et al. (1980) photosynthesis model was originally coupled to a stomatal model by Collatz et al. (1991; 1992). The coupled photosynthesis-conductance scheme was then integrated into the Simple Biosphere

Model developed by Sellers et al. (1992; 1996a, b, c, d). These initial applications were then built on by [Haxeltine and Prentice(1996a), Haxeltine and Prentice(1996b)].

We thank the reviewer for this suggestion to go back in time and explain the originals of this scheme. It certainly helps to trace the genesis of the modelling approach. In fact as [Pitman 2003] described it with the inclusion of the coupled photosynthesis-transpiration scheme the 3rd generation of Land Surface Models was formed. On top of that a second line of development which is briefly mentioned in [Pitman 2003], but not sufficiently explained therein, is the group of Dynamic Global Vegetation Models (DGVMs). Some DGVMs were developed to be coupled to LSMs and embedded in AOGCMs or Earth System Models, others by design to be stand-alone models to project climate impacts on the land biosphere, namely vegetation dynamics interacting with carbon, water and energy fluxes [Prentice et al.(2007)]. Many DGVMs also use the Farquhar-Collatz photosynthesis scheme which was developed further in Haxeltine and Prentice [Haxeltine and Prentice(1996a)] and then implemented in the BIOME-3 model [Haxeltine and Prentice(1996b)]. Since then more DGVMs have build up their photosynthesis schemes on those early publications so that today's DGVMs use this scheme to a large extent. Because a similar comment on providing a complete overview on the DGVMs using the Farquhar-Collatz photosynthesis scheme was made by Reviewer 1, we now added this overview in the newly added 2 paragraphs which review the respective literature on both lines of model development from line 33:

"The Farquhar-Collatz approach was implemented in the land surface of the SiB2 model by [Sellers et al.(1992), Sellers et al.(1996a)] where it replaced their empirical photosynthesis model. The photosynthesis model in SiB2 [Sellers et al.(1996b)] covers the co-limitation by Rubisco enzyme activity, light availability and export limitation of carbon compounds. Furthermore, it covers the gradient between inner-stomatal $CO_2$ concentration to the $CO_2$ concentration around the leaf surface in the computation of stomatal conductance. By implementing the semi-mechanistic photosynthesis model and coupling it to transpiration via stomatal conductance, the LSM could then not only investigate biophysical effects of climate change but also biogeochemical effects of rising atmospheric $CO_2$ in the Earth System [Pitman 2003]. The SiB2 model [Sellers et al.(1992), Sellers et al.(1996a)], the NCAR CCM2 model [Bonan et al. 1995], and the MOSES land surface model of the UK Met office [Cox et al. 1998] were among the first to implement this photosynthesis scheme and evaluated it against field campaigns. Today, the Farquhar-Collatz photosynthesis model is used in a number of Land surface models of the CMIP-5 Earth System Models, such as the Community Atmosphere Biosphere Land Exchange (CABLE) LSM of the Australian Community Climate Earth system Simulator (ACCESS, see [de Kauwe et al. 2015], and ref. therein) as well as the ORCHIDEE DGVM [Krinner et al.(2005)] of the IPSL-CM5 Earth System Model [Dufresne et al. 2013]. Different models of stomatal conductance were evaluated for the JS-BACH LSM [Reick et al. 2013] of the Max Planck Institute Earth System Model (MPI-ESM) to account for hydraulic properties and drought response [Knauer et al. 2015]. The Community Land Model CLM4.5 [Oleson et al. 2013] of the NCAR ESM use the Ball-Berry model of stomatal conductance and extended it to account for leaf temperature acclimation and leaf water potential [Bonan et al. 2014]; a similar approach was implemented in the JULES-vn5.6 land surface model [Oliver et al. 2022] of the UK Hadley Centre ESM [Sellar et al. 2019].

While Land surface models detail vertical water, energy and carbon profiles within the canopy, which extrapolates the photosynthetic capacity calculated at the leaf level to canopy photosynthesis [Sellers et al.(1996b)], stand-alone DGVMs often use a big-leaf approach and compute daytime photosynthesis for canopy conductance which goes back to the BIOME-3 model [Haxeltine and Prentice(1996b)] which opened up the second line of vegetation models by embedding the Farquhar-Collatz photosynthesis model in a modelling framework of plant physiology and vegetation dynamics in DGVMs [Prentice et al.(2007)]. The [Haxeltine and Prentice(1996b)] implementation is used in the LPJ model family originating from [Sitch et al. (2003)] and the LPJ-GUESS model [Smith et al. 2001, Smith et al. 2014], as well as the current LPJmLv4 model [Schaphoff et al.(2018a), Schaphoff et al.(2018b)]. Today, 14 DGVMs (stand-alone and coupled to land-surface models) contribute to the TRENDY intercomparison project

(https://blogs.exeter.ac.uk/trendy/) that informs the global carbon project on the state of the land carbon sink [Sitch et al. 2015]."

We compiled LPJmL using the -pg option to allow profiling. We executed the model for one grid cell to obtain the profile output from which the table on runtimes was produced using the gprof utility. The table contains the number of self calls and cumulative seconds as well as percentages of the runtime each routine required. It turned out that the photosynthesis routine using the bisection method required 38 per cent of the total computation time. The updated sentence now reads: "We quantified the runtime required by each submodule (or routine) of the LPJmL DGVM using the profiling option of the compilation command and the linux gprof utility. We found that the repeated execution of the photosynthesis routine demands a big fraction, i.e. 38%, of the computational time. All other routines require less than 11%."

We thank the reviewer for suggesting to provide such an overview in land surface schemes. We would have assumed that the exact numerical methods used would be documented in the peer-reviewed literature to provide a concise overview on the use of Newton's method in different modelling frameworks to solve coupled balance schemes. We were surprised to find very few additional references in the published literature. We searched the peer-reviewed data base Web of Science and also Google Scholar (using the keyword combination Farquhar AND photosynthesis AND Newton) and it seems these methods were rarely documented in the peer-reviewed literature. When working on the implementation of the Newton scheme for the photosynthesis, we found the hint in [Collatz et al. 1991], p.119, that the Newton method was used, but no documentation on the mathematical implementation, its computational cost or respective model code was provided. The same holds for [Pearcy et al. 1997] who looked at light regulation of two species at the leaf level. We found a description of photosynthesis model for rose leaf [Soo-Hyung and Lieth 2003], where the authors stated the use of the Newton-Raphson method to compute $\lambda$, but again no formulas or code were provided. In [Dubois et al. 2007] the statistical estimation of the parameters of the Farquhar-Collatz model is optimized by simultaneous estimation of multiple segments. For the required nonlinear regressions iterative methods like Gauss-Newton, steepest descent, or Levenberg-Marquardt algorithm are proposed. Again, there is no documentation. From the code in the supplements one can derive that Levenberg-Marquardt method was used. [Bonan et al. 2014] mentions numerical solution methods in their approach to include leaf water potentials, but again no details on this particular aspect are provided. This supports our view that the documentation and implementation of such a methods should be provided at least once.
We now refer to those references in the text: "Only a few, detailed specialized studies mention the use of Newton's or similar methods to solve coupled balance schemes, [Collatz et al. 1991, Pearcy et al. 1997, Soo-Hyung and Lieth 2003, Dubois et al. 2007], or extensions of the photosynthesis-transpiration scheme along the leaf-plant-soil continuum in DGVMs [Bonan et al. 2014] are mentioned, but none provide a documentation on the computational efficiency, or how the numerical method was implemented in the model and/or a code."

Lines 99-129, Section is difficult to follow without having the mathematical symbols defined at first use and the flow of the equations explained in narrative form. To improve readability, suggest defining each mathematical symbol in text at first use and also explaining what each

equation represents in physical terms rather than just presenting the mathematical derivation. We followed the suggestions of the referee and defined each symbol in the text (We still kept the table of symbols in the appendix). Each term of the defined function $f$ is now physically explained and some additional remarks were added that should make it easier to follow the computation of the derivative of $f$.

Lines 174-184, The argument developed here is a bit confusing. The lack of an impact of Newton's method on modeled pools and fluxes does not imply anything about the accuracy of the pool/flux calculations. The "accuracy of the photosynthesis scheme" must be defined relative to skill at explaining observations. Recent work by Walker et al. has highlighted the challenges in rigorously confronting the Farquhar-Collatz style schemes with observations due to the empirical coefficients that have been used as tuning knobs. One path forward is updating the current Farquhar-Collatz approach with the Johnson and Berry (2021) scheme which eliminates empirical coefficients, reduces the total number of free variables, and permits calculation of both gas-exchange and chlorophyll fluorescence.

Thanks for this valuable comment which helps to improve our manuscript substantially. Although it is possible to replace the Farquhar-Collatz scheme by the Johnson and Berry scheme [Johnson and Berry 2021], after intensive discussion we came to the conclusion that such an implementation into the LPJmL photosynthesis scheme is currently out of scope for this study. We now mention this step as a possible future development in our discussion section, were we state: "Future work on the photosynthesis approach could focus on the new Johnson and Berry scheme [Johnson and Berry 2021] with the advantage of calculating gas-exchange and relying less on empirical coefficients".

Instead, we have intensively studied the [Walker et al. 2020] paper and following their findings we have tested the influence of the following parameters wrt their sensitivity on GPP: $\theta, \alpha_{C3}, b_{C3}, k_{c25}, K_{o25}$ on changes to GPP. Although [Walker et al. 2020] have identified $V_{cmax}$ to be also a sensitive parameter in the photosynthesis scheme ([Walker et al. 2020], see Table 2 therein for $V_{cmax}$ parameter range), the way the Farquhar-Collatz approach is implemented in LPJmL does not allow to specify $V_{cmax}$ as a parameter. The LPJmL model computes $V_m$ as follows [Schaphoff et al.(2018a)], eq. (35):

$$V_m \quad = \quad \frac{1}{b_{C3}} \cdot \frac{c_1}{c_2} \cdot ((2\theta - 1) * s - (2\theta * s - c_2) * \sigma) \cdot APAR.$$

Therefore, the sensitivity of $V_{cmax}$ results from varying $b_{C3}$ indirectly since the reciprocal of $b_{C3}$ is used to calculate $V_{cmax}$ in a linear equation. Varying $b_{C3}$ is therefore the adequate sensitivity test which relates to $V_{cmax}$. We have now inserted the following text in the manuscript:
"In addition to improving the computational efficiency and numerical precision, parameter uncertainties have been tested by [Walker et al. 2020], who tested the sensitivity of $\theta, \alpha_{C3}, b_{C3}, k_{c25}, K_{o25}$ on their impacts on global GPP. The LPJmL model computes $V_m$ as follows [Schaphoff et al.(2018a)], eq. (35):

$$V_m \quad = \quad \frac{1}{b_{C3}} \cdot \frac{c_1}{c_2} \cdot ((2\theta - 1) * s - (2\theta * s - c_2) * \sigma) \cdot APAR.$$

Therefore, the sensitivity of $V_{cmax}$ results from varying $b_{C3}$ indirectly since the reciprocal of $b_{C3}$ is used to calculate $V_{cmax}$ in a linear equation. Varying $b_{C3}$ is therefore the adequate sensitivity test which relates to $V_{cmax}$. We varied each parameter by 10% independently and find that $\theta$ ($\alpha_{C3}, b_{C3}, k_{c25}, K_{o25}$) increases global annual GPP (AGPP, hereafter) by 1.67% (+6.69%, -1.67%, -0.35%, +0.14%). Table 1 shows the difference of the two most important parameter on global AGPP.

| parameter | $\Delta$ GPP relativ in % | $\Delta$ GPP absolut (GtC/yr) |
|---|---|---|
| $\theta$ | 1.67 | 2.384 |
| $\alpha_{C3}$ | 6.68 | 9.542 |
| $b_{C3}$ | -0.56 | -0.798 |
| $k_{c25}$ | -0.35 | -0.506 |
| $K_{o25}$ | 0.14 | 0.199 |

Table 1: Change in the AGPP after varying parameters by 10%.

Geographically, increasing $\theta$ yields higher AGPP mainly in the tropics and temperate forest regions, where AGPP increases up to 100 gC/m². However, AGPP increases between 200 and 500 gC/m² when changing $\alpha_{C3}$, see Fig.1. It turns out that AGPP is increased in all regions, where LPJmL simulates woody PFTs. Also here, largest effects are seen in (sub-)tropical and temperate regions which span larger areas than the areas with increased AGPP as a result of varying theta."

[Figure]

[Figure]

Figure R 1: Parameter sensitivity on Annual Gross Primary Productivity (AGPP) shown as the difference between the new parameter and the reference simulation. Both simulations have the Newton approach implemented. Increasing $\theta$ by 10 % increased AGPP mainly in forested regions (left panel). Increasing $\alpha_{C3}$ by 10 % has a much larger effect on AGPP, especially in the tropics (right panel).

We have now inserted the additional sensitivity test as a new paragraph, and include figure R 1 as the new Figure 4.

Figures D1-D12, The current figures simply summarize differences in model output across a variety of metrics; they add relatively little to the impact of the paper and it would be useful to distill them down to a smaller number of key visuals.

We understand that by showing the robustness of model simulations which are built on our benchmarking model evaluation system, we do not sufficiently display the main differences of the Newton approach and the parameter sensitivity. With the benchmarking we compare new model developments to a reference, i.e. master version. Because LPJmL has grown into a complex multi-sectorial model, we thought it to be important to show that the model is robust. We understand that this is not informative to the wider readership and show now only the figures related to GPP/NPP, vegetation carbon (i.e. biomass) and transpiration (because of the link via stomatal conductance) in terms of difference maps and only for NPP and transpiration as the global time series.

**References**

[revised manuscript text omitted]